# Patterns and correlates of active commuting in adults with type 2 diabetes: cross-sectional evidence from UK Biobank

Catherine L Falconer,[1,2] Ashley R Cooper,[2,3] Ellen Flint[4]

[1]South West Public Health, NHS Gloucestershire Foundation Trust, Cheltenham, UK
[2]NIHR Bristol Nutrition Biomedical Research Unit, University of Bristol, Bristol, UK
[3]Centre for Exercise, Nutrition and Health Sciences, University of Bristol, Bristol, UK
[4]Department of Social and Environmental Health Research, London School of Hygiene & Tropical Medicine, London, UK

**Correspondence to**
Professor Ashley R Cooper;
ashley.cooper@bristol.ac.uk

## ABSTRACT

**Objectives** To describe the active commuting (AC) patterns of adults with type 2 diabetes and how these relate to physical activity and sedentary behaviour in UK Biobank. Social and environmental correlates of AC will also be explored.

**Design** Cross-sectional analysis of a cohort study.

**Settings** This is a population cohort of over 500 000 people recruited from 22 centres across the UK. Participants aged between 37 and 73 years were recruited between 2006 and 2010.

**Participants** 6896 participants with a self-reported type 2 diabetes diagnosis who reported commuting to work and had complete covariate data were included in the analysis.

**Exposure measures** Exposure measures were AC to work, measured as usual mode of transport.

**Outcome measures** Outcome measures were weekly minutes of moderate to vigorous physical activity (MVPA), hours/day of sedentary time and participation in active travel.

**Results** AC (reporting walking or cycling to work only) was reported by 5.5% of participants, with the great majority using the car to commute (80%). AC was associated with an additional 73 (95% CI 10.8 to 134.9) and 105 (95% CI 41.7 to 167.2) weekly minutes of MVPA for men and women, respectively. AC was associated with reduced sedentary time ($\beta$ −1.1, 95% CI −1.6 to −0.7 hours/day for men; and $\beta$ −0.8, 95% CI −1.2 to −0.3 hours/day for women). Deprivation and distance from home to work were identified as correlates of AC behaviour.

**Conclusions** Rates of AC are very low in adults with type 2 diabetes. However, AC offers a potentially sustainable solution to increasing physical activity and reducing sedentary behaviour. Therefore, strategies to improve the environment and encourage AC may help to increase population levels of physical activity and reduce the disease burden associated with type 2 diabetes.

## Strengths and limitations of this study

► This study uses the UK Biobank to examine the active commuting behaviours of adults with type 2 diabetes. UK Biobank is a large, geographically diverse cohort that provides the largest exploration of active commuting in this population to date.

► UK Biobank provided the opportunity to be able to explore a multitude of employment, social and environmental factors that may contribute to active commuting.

► We were unable to distinguish between walking and cycling behaviour, and also were unable to explore different types of public transport modes, for example, bus or train.

► The analysis used in this study is cross-sectional and therefore we cannot infer causality.

► This study is further limited by the use of self-report measures of physical activity and sedentary time.

prevalence is 6%, and the ongoing treatment and management burden of T2DM and its associated complications place a considerable burden on the National Health Service.[4 5]

The role of physical activity (PA) in the prevention and management of T2DM is well understood.[6] However, fewer adults with T2DM achieve UK Government recommendations of at least 150 min moderate to vigorous physical activity (MVPA) per week,[7] and many interventions fail to achieve increases in PA sufficient to confer metabolic benefits.[7–9] Sedentary behaviour is also an independent risk factor for the development of T2DM.[10] Cross-sectional evidence from UK Biobank suggests that adults with T2DM are characterised by a phenotype of unhealthy behaviours, including prolonged sedentary time, physical inactivity and poor sleep, suggesting that these behaviours should be targeted in parallel.[11]

The role of active commuting (AC) in increasing PA and controlling excessive weight gain has been recognised and endorsed by

## INTRODUCTION

Type 2 diabetes mellitus (T2DM) is a disease associated with unhealthy lifestyle behaviours, such as poor diet and physical inactivity,[1] and displays a social gradient.[2] The worldwide age-standardised prevalence of T2DM has doubled to 8.5% since 1980.[3] In the UK the

the National Institute for Health and Care Excellence.[12] AC has the potential to increase individuals' PA and reduce sedentary time, and more widely to lower levels of traffic congestion and noise and air pollution,[13] and can be sustainably incorporated into daily living.[14] However, rates of AC are low in the UK. In the 2011 census, of the 58% of adults who responded to the question 'how do you usually travel to work? 67% reported private motorised transport as the principal mode of transport, 17.8% public transport, 10.9% walking and 3.1% commuting by bicycle.[15]

There is increasing evidence of an association between AC and measures of obesity.[16 17] In the UK Understanding Society cohort, AC has been shown to be predictive of lower body mass index (BMI) and body fat percentage, and a lower adjusted odds of reporting a diabetes diagnosis compared with car users.[16 18] Similarly in UK Biobank, AC was significantly and independently associated with lower BMI and body fat percentage.[17] The protective effect of AC is likely to be mediated through increased PA, as demonstrated in a UK study that used accelerometry and Global Positioning System (GPS) receivers to show that walking to work contributed 47.3% of daily MVPA.[19]

Causal evidence from longitudinal or experimental studies is rare. A longitudinal analysis of the UK Biobank cohort looked at changes in BMI in response to changes in commuting mode over a 4-year follow-up. Individuals who transitioned from car commuting at baseline to active or public transportation modes at follow-up had a decrease in BMI of $-0.30\,kg/m^2$ (95% CI $-0.47$ to $-0.13$).[20]

Promoting AC may help to increase population levels of activity; however, there is a need to understand the correlates of AC to develop appropriate interventions. Some key determinants of commuter cycling and walking behaviour have been identified as high population density, dedicated cycling paths, short trip distances and good traffic and road conditions, such as the presence of traffic calming measures.[21 22] Most work to date has been performed in healthy adults, and there is a need to understand how these findings relate to populations with T2DM who are characterised by obesity and physical inactivity.

The primary aim of this work is to describe the AC patterns of adults with T2DM in UK Biobank. Specifically, the study explores how AC is associated with PA and sedentary time in this population (objective 1). The study will also aim to identify social and environmental factors (age, social class and traffic density) that may predict participation in AC (objective 2).

## METHODS
### Study population
Baseline data from the UK Biobank study were used (project 19307). UK Biobank is a large population study that aims to examine how the environment, lifestyle and genes interact to affect health.[23] Full recruitment

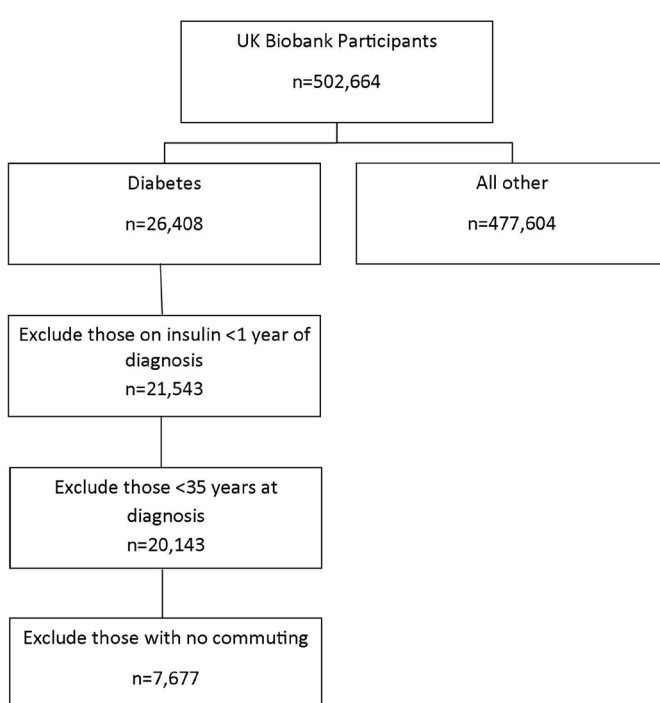

**Figure 1** Flow chart demonstrating how the population sample was identified.

procedures are described in detail elsewhere[24] and briefly summarised in figure 1.

Between 2006 and 2010, recruitment invitations were sent to 9.2 million people from 22 assessment centres around the UK to obtain a sample of 502 565 adults aged 37–73 years.[11] Following recruitment, participants provided written informed consent and attended an assessment centre for data collection. A touchscreen questionnaire covered sociodemographics, occupation, lifestyle, early exposures, cognitive function, family history and medical history. Participants also underwent a verbal interview and physical measures, and provided a blood/urine sample. All data were anonymised. The UK Biobank Access Subcommittee granted approval for the data access under a Material Transfer Agreement with the London School of Hygiene & Tropical Medicine (LSHTM), which provided ethical approval for this project (LSHTM ethics reference: 10838).

### Study sample
Participants who reported having diabetes or T2DM were selected (n=26 408). We excluded anyone <35 years at diagnosis of diabetes and who reported taking insulin within the first year of diagnosis to reduce the likelihood of type 1 or monogenic forms of diabetes[11] (n=20 143). Only participants who reported commuting behaviours and had complete covariate data were included in the analysis (n=6896).

### AC measures
In response to the question 'what types of transport do you use to get to and from work?' participants could select one or more of the following options: car/motor

vehicle, walk, public transport, cycle. In order to answer the two study objectives, two distinct exposure variables were derived: (1) a three-category exposure variable: (1) *car* or *car and public transport*; (2) *public transport* and *mixed* (car and walk/cycle; public transport; and walk/cycle); (3) AC: *walk* or *cycle* only; and (2) a binary variable: (1) no AC and (2) AC (walk or cycle only). Only participants who exclusively reported walking or cycling behaviour were included in the AC categories.

### Physical activity

PA was assessed using questions adapted from the validated short form of the International Physical Activity Questionnaire (IPAQ).[25] The questions asked participants to self-report the frequency, duration and intensity of walking, moderate and vigorous activity. From these questions, we derived three measures of PA. A variable for total PA (total metabolic equivalent (MET)-min/week) was derived by weighing time spent in vigorous, moderate and walking activity by the relevant energy expended for each activity to produce MET-min/week of PA.[26] Duration of MVPA was derived as the sum of reported duration and frequency of moderate and vigorous activity. From this, a variable for 'attainment of physical activity guidelines' (150 min of moderate PA or 75 min of vigorous activity per week) was derived.[7]

### Sedentary behaviour

Self-reported television viewing, computer use and driving behaviour were collected from the touchscreen questionnaire. An estimate of average total sedentary time was derived by summing the daily reported duration of television viewing, computer use and driving. Total sedentary time is expressed as hours per day.

### Environmental correlates

Local environmental exposure variables were collated by the Small Area Health Statistics Unit[27] as part of the BioSHaRE-EU Environmental Determinants of Health Project.[28] Air pollution estimates were modelled for the year 2010 using a land use regression (LUR) model developed as part of the European Study of Cohorts for Air Pollution Effects (ESCAPE).[29] ESCAPE estimates of particulate matter (PM) are valid up to 400 km from the monitoring area, which in this case was Greater London. Therefore, participants who have an address more than 400 km from this site were not assigned a value for PM2.5–10. Traffic variables, including inverse distance to a major road and traffic density, were also calculated using the LUR model, based on Eurostreets (V.3.1) digital road network for the year 2008. Traffic density is measured as the average total number of motor vehicles per 24 hours on the nearest road. These correlates were selected on the basis that they were available on an individual level and may help to inform further analysis of environmental correlates of commuting behaviour.

### Covariates

A range of confounding variables were considered. Socioeconomic covariates were Townsend Deprivation Index of residential area, annual gross household income and highest educational qualification. Further demographic covariates were age (years), sex, ethnicity and census-derived residential area population density classification (urban/fringe, rural). The potential confounding effects of occupational PA were considered: job involves standing/walking; job involves heavy manual/physical work; shift work; non-work related transport modes (active modes: yes/no). Health covariates were poor self-rated health (yes/no). Commute frequency (number of outward journeys per week, truncated at 10) and distance between home and workplace (reported in miles) were included as covariates in the model.

### Statistical analyses

Descriptive analyses were undertaken to describe the commuting patterns of the population, the prevalence of PA and sedentary behaviours, and the patterning of hypothesised confounding variables. Continuous variables are displayed as mean and SD, and frequencies are used for categorical variables.

To address objective 1, multivariate linear regression was used to describe the association between AC and each of the two outcome variables (PA and sedentary time) adjusting for potential confounding variables in two stepwise series of three nested models. The first model presents the unadjusted association between the exposure (commuting mode) and outcome of interest. The second model adjusts for demographic and socioeconomic factors. The third model additionally adjusts for employment-related confounders such as distance from home to work.

To address objective 2, multivariate logistic regression modelling was used to explore potential sociodemographic, environmental and employment-related correlates of AC behaviour. The model used was adjusted for all included variables. All analyses controlled for geographical clustering by assessment centre, and potential interactions by gender, ethnicity and household income were explored. For these analyses, the outcome was expressed as a binary categorical variable ((1) not AC and (2) AC). All analyses were conducted using Stata V.14.[30]

## RESULTS

The final sample was 6896 participants with T2DM who reported commuting behaviour and provided complete data on all covariates (table 1).

Car travel was reported by 83.2% of male and 75.2% of female participants, with walking or cycling exclusively reported by 4.5% of male and 7.2% of female participants. Women spent less time sedentary than men (4.9±2.7 hours/day vs 6.0±3.1 hours/day), while men reported more weekly MVPA (263.9±446.4 min/week vs 214.9±383.4 min/week).

The results of analyses to investigate the association between commuting mode, PA and sedentary behaviour

**Table 1** Demographic, commuting behaviour, employment, physical activity and environmental characteristics of the study population

| Variable | Male (n=4473) | Female (n=2423) | Total (n=6896) |
|---|---|---|---|
| Demographics | | | |
| Age (years) | 56.7±6.5 | 54.53±6.6 | 55.9±6.6 |
| Ethnicity (%) | | | |
| White British | 81.8 | 78.2 | 80.5 |
| Other white | 5.3 | 6.5 | 5.7 |
| South Asian | 6.2 | 4.2 | 5.5 |
| Black Caribbean | 1.3 | 3.9 | 2.2 |
| Black African | 1.8 | 2.1 | 1.9 |
| Chinese | 0.3 | 0.6 | 0.4 |
| Mixed ethnicity | 0.6 | 0.8 | 0.7 |
| Other ethnicity | 2.7 | 3.8 | 3.1 |
| Townsend Deprivation Index (quintiles) (%) | | | |
| 1 (most deprived) | 16.1 | 14.1 | 15.4 |
| 2 | 17.2 | 15.8 | 16.7 |
| 3 | 19.6 | 20.9 | 20.1 |
| 4 | 21.2 | 22.1 | 21.8 |
| 5 (least deprived) | 25.5 | 27.1 | 26 |
| Highest educational qualification (%) | | | |
| University | 29.6 | 30.8 | 30 |
| A levels/AS levels | 11.2 | 11.2 | 11.4 |
| GCSE/O levels | 20.5 | 25 | 22.1 |
| CSEs or equivalent | 6.4 | 7.9 | 6.9 |
| NVQ or HND or HNC | 11.5 | 7 | 9.9 |
| Other professional qualifications | 5.32 | 6.2 | 5.6 |
| None | 15.6 | 11.6 | 14.2 |
| Average household income (%) | | | |
| <£18 000 | 12.2 | 20.5 | 15.1 |
| £18 000–30 999 | 27.4 | 28 | 27.6 |
| £31 000–£51 999 | 30.9 | 29.3 | 30.4 |
| £52 000–£100 000 | 23.8 | 18.6 | 22 |
| >£100 000 | 5.7 | 3.5 | 4.9 |
| Poor self-rated health (%) | 9.9 | 9 | 9.6 |
| Employment | | | |
| Job involves heavy lifting (%) | | | |
| Never/rarely | 59.8 | 65.8 | 61.9 |
| Sometimes | 25 | 22.5 | 24.1 |
| Usually | 8.1 | 5.7 | 7.2 |
| Always | 7.1 | 6 | 6.7 |
| Job involves walking (%) | | | |
| Never/rarely | 31.3 | 32.8 | 31.8 |
| Sometimes | 32.7 | 31.2 | 32.2 |
| Usually | 16.1 | 13.8 | 15.3 |
| Always | 19.9 | 22.2 | 20.7 |
| Shift work | 7.7 | 5.9 | 7.1 |

| Table 1 Continued | | | |
|---|---|---|---|
| **Variable** | **Male (n=4473)** | **Female (n=2423)** | **Total (n=6896)** |
| Commuting behaviour | | | |
| Commute frequency (number of outward journeys/week) | 4.6±2.1 | 4.3±1.8 | 4.5±2.0 |
| Distance from home to work (miles) | 14.9±28.4 | 7.5±9.9 | 12.3±24.0 |
| Commuting mode (%) | | | |
| Motorised travel | 83.2 | 75.3 | 80.4 |
| Motorised travel plus active travel | 12.3 | 17.5 | 14.1 |
| Active commuting | 4.5 | 7.2 | 5.4 |
| Non-work active travel | 40.4 | 40.2 | 40.3 |
| Physical activity and sedentary time | | | |
| Daily sedentary time (hours) | 6.0±3.1 | 4.9±2.7 | 5.6±3.0 |
| Total weekly MVPA (min) | 263.9±446.4 | 214.9±383.4 | 246.6±425.9 |
| Total weekly MET*-min | 2373.3±3099.8 | 2014.7±2854.7 | 2245.5±3106.2 |
| Achieves PA guidelines† (%) | 54.3 | 53.8 | 54.1 |
| Environmental characteristics | | | |
| Close to major road (%) | 8 | 8.2 | 8.1 |
| Population density (urban/fringe) (%) | 94.5 | 95.7 | 94.9 |
| Inverse distance to nearest road (1/m) | 0.05 (0.05, 0.05) | 0.05 (0.05, 0.05) | 0.05 (0.05, 0.05) |
| Air pollution PM2.5–10 (µg/m$^3$) | 6.5±0.9 | 6.5±0.9 | 6.5±0.9 |
| Traffic density (vehicles/day) | 1699.4±5663.6 | 1716.2±6057.0 | 1698.6±5663.6 |

*MET, metabolic equivalents (1 MET=1 kcal/kg/hour).

†PA guidelines, physical activity guidelines (150 min MVPA/week or 75 min vigorous activity/week).[7]

AS, Advanced Subsidiary; GCSE, General Certificate of Secondary Education; NVQ, National Vocational Qualification; HND, Higher National Diploma; HNC, Higher National Certificate; MVPA, moderate to vigorous physical activity; PM, particulate matter.

(objective 1) are displayed in table 2. A significant interaction for gender was found so stratified analyses are presented (p<0.1).

In all models, AC was associated with higher MVPA. In fully adjusted models, compared with motorised travel, AC was associated with an additional 72.9 (95% CI 10.8 to 134.9) min/week of MVPA for men and 104.5 (95% CI 41.7 to 167.2) min/week for women. The association with motorised travel and AC was less clear. In men, there was a suggestion that motorised travel and AC were associated with less MVPA compared with motorised travel alone. This association was attenuated once environmental factors were included in the model (adjusted β 29.6, 95% CI –70.6 to 11.4).

AC was also consistently associated with increased total PA in both genders, an effect that survived adjustment for all confounding factors (β 586.6, 95% CI 150.9 to 1022.3 for men; and β 722.3, 95% CI 270.3 to 1174.1 for women). In unadjusted models, motorised travel and AC were associated with increased total PA for women only; however, this was attenuated once employment and demographic factors were included in the model.

In both men and women, and in all models, consistent associations were seen between commuting mode and total sedentary time. In fully adjusted models, compared with male motorised travellers, AC was associated with 1.1 (95% CI –1.6 to –0.7) fewer hours/day sedentary, while

mixed motorised travel and AC modes were associated with 2.2 (95% CI 2.5 to 1.9) fewer sedentary hours/day. A similar effect was seen in women with AC associated with 0.8 (95% CI –1.2 to –0.3) fewer sedentary hours/day, and mixed motorised travel and AC were associated with 1.1 (95% CI 1.5 to 0.8) fewer sedentary hours/day compared with car users.

The results of multivariate logistic regression performed to identify social, employment and environmental correlates of AC behaviour (objective 2) are displayed in table 3. There was evidence of a significant interaction for gender so stratified results are presented.

In model 1 for both men and women, the odds of participating in AC increased with increasing neighbourhood deprivation score, with the strongest effect seen in deprivation quintile 5 compared with quintile 1 (adjusted OR: 4.24, 95% CI 2.41 to 7.44 for men; and adjusted OR: 3.34, 95% CI 1.61 to 6.94 for women). In fully adjusted models, no other social factors were associated with AC behaviour.

Model 2 additionally adjusted for employment-related factors. In women, neighbourhood deprivation remained a significant correlate of AC; however, for men the association was attenuated. Distance to work was the strongest correlate of AC. In men, the odds of AC were 88% lower (adjusted OR: 0.12, 95% CI 0.08 to 0.17) in those who lived between 1.5 and 4 miles from work compared

**Table 2** Results of linear regression modelling to investigate the associations between commuting behaviour, physical activity and sedentary time, stratified by gender

| | MVPA (min/week)* | | Total MET-min/week | | Total sedentary time (hours/day) | |
|---|---|---|---|---|---|---|
| | Male β (95% CI) | Female β (95% CI) | Male β (95% CI) | Female β (95% CI) | Male β (95% CI) | Female β (95% CI) |
| Car (reference) | 0.0 | 0.0 | 0.0 | 0.0 | 0.0 | 0.0 |
| MT/AC† | −49.9 (−89.8 to 10.1) | 12.5 (−27.9 to 52.8) | 93.0 (−381.0 to 193.0) | 312.3 (12.3 to 612.4) | −2.0 (−2.3 to 1.7) | −1.0 (−1.3 to 0.7) |
| Active commuting | 86.4 (23.1 to 149.7) | 156.2 (97.0 to 215.4) | 777.0 (320.6 to 1233.3) | 1287.3 (847.3 to 1727.4) | −1.3 (−1.8 to 0.9) | −1.0 (−1.4 to 0.6) |
| Model 2‡ | | | | | | |
| Car (reference) | 0.0 | 0.0 | 0.0 | 0.0 | 0.0 | 0.0 |
| MT/AC† | −57.4 (−98.6 to 16.2) | −3.0 (−45.9 to 40.1) | −156.1 (−452.4 to 140.3) | 155.2 (−163.4 to 473.9) | −2.2 (−2.4 to 1.9) | −1.2 (−1.5 to 0.9) |
| Active commuting | 74.6 (10.9 to 138.4) | 145.3 (85.5 to 205.2) | 665.5 (207.0 to 1124.1) | 1149.4 (706.3 to 1592.5) | −1.5 (−1.9 to 1.0) | −1.1 (−1.5 to 0.7) |
| Car (reference) | 0.0 | 0.0 | 0.0 | 0.0 | 0.0 | 0.0 |
| MT/AC† | −29.6 (−70.6 to 11.4) | 11.5 (−35.3 to 58.3) | 47.0 (−240.9 to 334.9) | 290.8 (−46.0 to 627.6) | −2.2 (−2.5 to 1.9) | −1.1 (−1.5 to 0.8) |
| Active commuting | 72.9 (10.8 to 134.9) | 104.5 (41.7 to 167.2) | 586.6 (150.9 to 1022.3) | 722.3 (270.3 to 1174.1) | −1.1 (−1.6 to 0.7) | −0.8 (−1.2 to 0.3) |

*MVPA: Moderate to vigorous physical activity.
†MT/AC: motorised travel and active commuting.
‡Model is adjusted for age, ethnicity, average household income, Townsend deprivation score, assessment centre, poor health.
§Additionally adjusted for distance from home to work, commute frequency, population density, non-work active travel and job involves walking. Model 1 presents unadjusted associations.
MET, metabolic equivalents.

**Table 3** Results of logistic regression modelling to identify potential social, employment and environmental correlates of active commuting

| | Model 1 Individual factors | | Model 2 Employment factors | | Model 3 Environmental factors | |
|---|---|---|---|---|---|---|
| | Male OR (95% CI) | Female OR (95% CI) | Male OR (95% CI) | Female OR (95% CI) | Male OR (95% CI) | Female OR (95% CI) |
| Social factors | | | | | | |
| Age (years) | 1.0 (0.9 to 1.0) | 0.98 (0.95 to 1.00) | 0.98 (0.96 to 1.00) | 0.97 (0.94 to 0.99) | 1.01 (0.98 to 1.03) | 0.97 (0.95 to 0.99) |
| Townsend Deprivation Index quintile | | | | | | |
| 1 (ref) | 1.0 | 1.0 | 1.0 | 1.0 | 1.0 | 1.0 |
| 2 | 1.13 (0.58 to 2.22) | 2.09 (0.93 to 4.71) | 0.88 (0.42 to 1.84) | 2.76 (1.10 to 6.92) | 1.19 (0.60 to 2.37) | 1.99 (0.88 to 4.50) |
| 3 | 1.90 (1.03 to 3.49) | 3.00 (1.42 to 6.37) | 1.35 (0.69 to 2.64) | 2.54 (1.10 to 5.90) | 1.80 (0.96 to 3.37) | 2.69 (1.26 to 5.75) |
| 4 | 2.04 (1.12 to 3.71) | 3.25 (1.55 to 6.83) | 1.49 (0.77 to 2.88) | 2.60 (1.14 to 5.95) | 1.79 (0.96 to 3.34) | 2.92 (1.38 to 6.19) |
| 5 | 4.24 (2.41 to 7.44) | 3.34 (1.61 to 6.94) | 2.28 (1.23 to 4.21) | 3.07 (1.36 to 6.94) | 3.32 (1.84 to 6.01) | 3.08 (1.47 to 6.48) |
| Average household income (£) | | | | | | |
| <£18 000 (ref) | 1.0 | 1.0 | 1.0 | 1.0 | 1.0 | 1.0 |
| £18 000–£30 999 | 0.68 (0.44 to 1.04) | 0.72 (0.48 to 1.09) | 0.97 (0.59 to 1.59) | 1.25 (0.77 to 2.04) | 0.61 (0.39 to 0.97) | 0.76 (0.50 to 1.16) |
| £31 000–£51 999 | 0.69 (0.44 to 1.07) | 0.39 (0.24 to 0.62) | 1.23 (0.74 to 2.04) | 0.94 (0.53 to 1.65) | 0.65 (0.42 to 1.03) | 0.41 (0.25 to 0.67) |
| £52 000–£100 000 | 0.85 (0.53 to 1.37) | 0.35 (0.20 to 0.62) | 1.34 (0.77 to 2.33) | 0.88 (0.45 to 1.74) | 0.80 (0.49 to 1.32) | 0.37 (0.21 to 0.67) |
| >£100 000 | 1.10 (0.58 to 2.11) | 0.90 (0.40 to 2.01) | 1.27 (0.59 to 2.79) | 4.27 (1.47 to 12.40) | 0.90 (0.43 to 1.89) | 0.75 (0.30 to 1.87) |
| Assessment centre | 0.97 (0.95 to 1.00) | 1.00 (0.98 to 1.03) | 0.98 (0.95 to 1.01) | 1.02 (0.98 to 1.05) | 0.98 (0.95 to 1.00) | 1.00 (0.97 to 1.03) |
| Poor health | 0.70 (0.41 to 1.18) | 0.83 (0.47 to 1.46) | 0.56 (0.31 to 1.00) | 1.04 (0.53 to 2.04) | 0.62 (0.35 to 1.11) | 0.89 (0.50 to 1.56) |
| Employment characteristics | | | | | | |
| Distance from home to work (miles) | | | | | | |
| <1.5 (ref) | | | 1.0 | 1.0 | | |
| 1.5–4 | | | 0.12 (0.08 to 0.17) | 0.07 (0.05 to 0.11) | | |
| 4–10 | | | 0.01 (0.01 to 0.03) | 0.01 (0.05 to 0.03) | | |
| >10 | | | 0.002 (0.001 to 0.01) | 0.00 | | |
| Commute frequency | | | 0.95 (0.86 to 1.05) | 1.01 (0.91 to 1.14) | | |
| Job involves walking | | | 0.97 (0.83 to 1.13) | 1.20 (1.00 to 1.43) | | |
| Shift work | | | 0.85 (0.43 to 1.67) | 0.76 (0.31 to 1.88) | | |
| Environmental characteristics | | | | | | |
| Population density (ref: urban) | | | | | 0.12 (0.02 to 0.90) | 0.39 (0.09 to 1.62) |
| Air pollution (PM10 µg/m³) | | | | | 1.01 (0.84 to 1.21) | 0.89 (0.72 to 1.09) |
| Traffic density (number of cars) | | | | | 1.0 (0.99 to 1.0) | 0.99 (0.99 to 1.00) |
| Close to major road (ref: yes) | | | | | 1.48 (0.76 to 2.89) | 1.41 (0.64 to 3.09) |
| Distance to nearest road (1/mile) | | | | | 0.15 (0.01 to 4.15) | 0.49 (0.02 to 9.67) |

PM, particulate matter.

with those who lived within 1.5 miles. The odds of AC were 93% lower (adjusted OR: 0.07, 95% CI 0.05 to 0.11) in women who lived between 1.5 and 4 miles from work, with no participants who lived further than 10 miles away reporting AC. In men, no other employment characteristics were associated with AC. In women, there was a suggestion that women whose job involved walking were more likely to use AC (adjust OR: 1.20, 95% CI 1.00 to 1.43).

Model 3 explored possible environmental correlates of AC behaviour. The only variable associated with AC was population density, in men only. The men who lived in a rural area had a 78% reduced odds of AC compared with those who lived in an urban or fringe area (adjusted OR: 0.12, 95% CI 0.02 to 0.90). No other environmental variables were associated with AC.

## DISCUSSION

A substantial majority (80%) of adults with T2DM reported commuting by car. AC was reported by 5.5% of the sample, of whom less than 1% reported cycling. AC was strongly and consistently associated with higher PA, whether expressed as MVPA or total MET-min/week. In fully adjusted models, AC was associated with an additional 73 min/week of MVPA for men and 105 min/week of MVPA for women, making a substantial contribution to meeting the UK Government recommended levels of 150 min/week MVPA.[7] This finding supports previous work by Audrey et al,[19] which found levels of PA to be higher in those who walked to work compared with those who did not. Furthermore the association with MVPA seen in the present study is substantially larger than achieved in a similar population undergoing an intensive PA intervention,[8] suggesting that adoption of AC could be an effective intervention for people with T2D.

The importance of sedentary time for both prevention and management of type 2 diabetes has gained research prominence in recent years,[31 32] and we found a substantial association between AC and lower sedentary time. In both sexes both AC and public and mixed modes of transport were associated with over an hour less sedentary time per day compared with car use. Since adults with T2DM typically spend less than 3% of the day engaged in MVPA, interventions that focus on reducing sedentary time may be more effective to improve health[33]; reallocating 30 min of prolonged sedentary time to more active pursuits can improve body composition and cholesterol levels in adults with T2DM.[34]

The predominant correlate of AC was distance to work, with few people who lived over 1.5 miles from their workplace choosing AC and those living in rural areas less likely to actively commute. Although distance from home to work is not modifiable, AC as the whole or part of the journey could be encouraged by providing more cycleways, by road safety measures to limit traffic speed and improving access to parks and green spaces.[35 36] We were unable to examine the association of access to transport

as a correlate of AC, but it can be hypothesised this would correlate strongly with the transport method chosen. Furthermore, we used a crude measure of AC (walking or cycling only) and were therefore unable to explore the role of AC as a contributor to the overall journey. Individuals who live beyond 3–6 miles from work are unlikely to solely use AC modes to travel to work and are more likely to combine AC with another form of activity, such as the car or public transport. Therefore the role of AC as a component of the overall journey should be explored in future research.

In the current study, the results for public transport were equivocal with no apparent observations with PA observed. This contrasts with previous research[17] and may be due to an inability to differentiate between the different types of public transport taken and the public transport category also including people who reported mixed modes of motorised travel and AC. There are likely to be variations in the energy requirements of different transport modes, which should be examined. It could also be hypothesised that walking or activity behaviours associated with public transport are difficult to recall and record in the IPAQ. However, in previous research public transport has been shown to contribute to PA and therefore warrants further investigation.

The final aim of the study was to explore the social and environmental correlates of AC behaviour. Those who lived in a rural area were less likely to use active modes of commuting, but no other environmental variables, such as traffic density, were shown to be associated. This may be due to an overall lack of variation in the environmental variables and suggests a more diverse sample may be required. For example, there is a lack of interindividual variation in the measure of air pollution, which may explain the lack of association seen.

This study has some major strengths. UK Biobank provides a large, geographically diverse data source allowing multiple social, environmental and employment related factors to be adjusted for in analyses. The study sample is restricted to adults with T2DM and is the first study of its kind to explore these behaviours on such a scale.

The study also has some important limitations. The analyses were cross-sectional and therefore causality cannot be determined. The small numbers of participants reporting AC did not allow walking and cycling to be examined separately. The relative energy demands of these behaviours differ, and previous research has suggested these behaviours are performed by different portions of society.[37] The current study was also unable to distinguish between the different modes of public transport. The observed gender difference suggests there may be differences in the work and travel patterns of men and women in this study population. The average distance from home to work was almost twice as far for men, which may explain why more women reported active modes of travel. The study is further limited by self-report measures of PA and sedentary behaviour.

The study sample is predominantly of white British ethnicity, which does not reflect the distribution of type 2 diabetes in the UK, limiting the generalisability of the findings. The sample is also limited to those who work, and therefore can only examine the role of AC on PA and sedentary time. The contribution of non-work active travel may be of more interest for people with T2DM who in general are older than the UK average.

In conclusion, AC is associated with increased PA and reduced sedentary time in people with type 2 diabetes. This may have a beneficial effect on glucose control and could contribute towards reduced treatment costs. Strategies to increase AC in adults with T2DM may increase levels of PA, and thus contribute to reducing the burden of this disease.

**Contributors** CLF, ARC and EF made substantial contributions to the conception and design, acquisition of data, or analysis and interpretation of data. All authors drafted the article or revised it critically for important intellectual content. All authors approved the final version of the manuscript to be published.

**Funding** This project was funded by the National Institute for Health Research Bristol Nutrition Biomedical Research Unit. The funding covered the cost of data access only.

**Disclaimer** This research has been conducted using the UK Biobank Resource under Application Number 19307.

**Competing interests** None declared.

**Ethics approval** UK Biobank and London School of Hygiene & Tropical Medicine ethics board.

**Provenance and peer review** Not commissioned; externally peer reviewed.

**Data sharing statement** No additional data are available.

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
