## [Reviewer comments · BMJ Open]

ARTICLE DETAILS

TITLE (PROVISIONAL)	Patterns and correlates of active commuting in adults with type 2 diabetes: cross-sectional evidence from UK Biobank
AUTHORS	Falconer, Catherine; Cooper, Ashley; Flint, Ellen

VERSION 1 - REVIEW

REVIEWER	Erica Hinckson Auckland University of Technology
REVIEW RETURNED	19-Apr-2017

GENERAL COMMENTS	1. Please explain the age range why not younger than 35? Data indicate that the incidence of type 2 can be seen at earlier2. Environmental characteristics did not include access to transport, very important feature for active commuting3. The differentiation of active commuting measures may need further explanation. How do these differ? (1) car or car and public transport; why have these two been merged together? (2) public transport and mixed; what does mixed include? and where is the option of car with cycling/walking?
---

REVIEWER	Oliver Mytton MRC Epidemiology Unit, University of Cambridge
REVIEW RETURNED	21-Apr-2017

GENERAL COMMENTS	Abstract Please add "UK Biobank" Participant line, clarify type 2 DM The abbreviation for AC is not spelt out in the abstract not given; do you want a line for "exposure measures"? Introduction 1) While the introduction is accurate, the case for why the work has been done could be much clearer. It may also benefit from being a bit shorter and more focused. BMI is relevant (particularly to aetiology of diabetes), but does it require two paragraphs? My personal view is that the strongest argument for undertaking this work is that people with T2DM tend to be less active, and increasing physical active can be an important part of their treatment. AT is likely to be a sustainable way for people to maintain relatively high levels of PA – the aim of this work was to understand whether (in
--

people with T2DM) AT was associated with increased PA; and further to understand the correlates of AT (with a view to designing interventions).

2) The shift from active travel to active commuting (with paragraph 3) is not explained. They are different.

3) First line, the phrase “driven by” – may imply causes, and given this study is about people *with* diabetes, it is not informative in terms of aetiology

4) 2nd para – what proportion of patients with T2DM meet guidelines according to this data source, in your sample the prevalence looks quite high (>50%) with on average well over 150 minutes per week.

5) Line 34, suggest add “traffic” to clarify

6) Ref 15 – are these responses based on ‘usual’ mode of travel (i.e. a single choice is forced)? Would be helpful to clarify

7) Line 41 – it is not just **cross-sectional** evidence, you later cite longitudinal, perhaps observational

8) Page 5, line 15 – what are good traffic conditions, do you mean low traffic volumes? Or slow speed?

9) Page 5, Lines 11-18, is there any reason why these findings might be different in people with T2DM?

Methods

10) Line 44, please insert a space after 9.2

11) Study sample, is it worth considering a flow diagram (item 13c on the STROBE statement)?

12) Active commuting measures – it might be helpful to spell out, perhaps in a table, what these different groups are (i.e. which responses go into which groups), for those who are less familiar with the questionnaire. Would labels like: motorised travel; motorised travel plus active travel; active travel only be more descriptive?

13) Physical activity; I wonder whether a statement along the lines of “From these questions, we derived three measures of physical activity...”

14) Is total physical activity the best or usual term – it is a measure of physical activity energy expenditure (i.e. product of duration and intensity). Total PA doesn't quite feel right because you have only summed moderate, vigorous and walking (with walking often being moderate in intensity or close to)

15) Line 35 – should the units be MET-minutes (not MET/minutes)?

16) MVPA – would it be clearer to call this duration of MVPA (in part to distinguish from total PA)

17) Environmental correlates. Why were these measures chosen? Many of the measures seem sensible to consider, but I am less convinced about air pollution (of which people are largely unaware and may be correlated with other measures like density)

Table 1

18) For deprivation, could you please indicate the most and least deprived quintile

19) Most of the numbers are % - could this be indicated please

20) Commute frequency – per week?

21) Distance for home – mean or median?

22) Please clarify what traffic density means; does this refer to the road on which the person lives?

Table 2

23) MVPA, suggest add label (minutes)?

24) MET-minutes or MET minutes; suggest should be consistent with text; and is this per week or per day?

25) Total sedentary time, please indicate is this per day?

26) Could you please add the details of the models as a foot note to

	the table, i.e. the variables you Table 3 27) Why did you not adjust of employment and environment characteristics in a single model? Results 28) Page 11, para 3: the values quoted are positive but the values given in bracket (as the 95% CI are negative), please review 29) Page 11, line 36 – Table *3* 30) Line 37, suggest add the p-value for the interaction term 31) Page 12, line 54 – “overwhelming” feels like an odd word; some other variables still remain significant so whilst distance has a very strong association, other associations do still persist Discussion 32) Line 25, “higher PA” – please clarify MVPA or total PA 33) Line 32, effect size or association. Observation studies (particularly cross-sectional ones), cannot demonstrate an effect. 34) Lines 55-59; I do not see how these measures can overcome the problem of a long commuting distance. 35) Page 14, line 1; could you elaborate a little further on why public transport use was associated with little or no effect on MVPA? Are people who undertake PT unlikely or not able report some of the walking associated with PT in the questionnaire used? Did you look at the descriptive stats by commuting category? In analyses I have done, those who use the public transport are much more likely to live further from home and/or have long duration of commutes (and thus it seems plausible that they may have to reduce other activities to compensate – e.g. recreational MVPA). For this reason comparing public transport users (depending on the characteristics of your sample) with car users may not be a ‘fair’ comparison if car commuters live closer to work. Conclusion This might benefit from being more closely aligned to the study. I would consider dropping the statement “Furthermore, AC has the dual benefit...”. Something along the lines of “strategies to increase active commuting in adults with T2DM may increase levels of physical activity, and thus contribute to reducing the burden of this disease”, feels more appropriate. Improve the environment sounds too vague – and your work only looks at correlates.
--	---

VERSION 1 – AUTHOR RESPONSE

Reviewer: 1
Comments

The authors examined patterns and correlates of active commuting in adults with type 2 diabetes. This is a very interesting topic and the research is worth publishing. The manuscript is written well and all components of the research addressed appropriately. I found a couple of things confusing. Please provide justification for the following:

1. Please explain the age range why not younger than 35? Data indicate that the incidence of type 2 can be seen at earlier

The age range included in the current study reflects that of the UK Biobank cohort. UK Biobank originally recruited adults aged 40 to 69 years to the cohort. We acknowledge that the incidence of type 2 diabetes is increasing in adults below the age of 35 years old; however, these individuals were

not recruited to UK Biobank.

2. Environmental characteristics did not include access to transport, very important feature for active commuting

We were limited in the environmental characteristics that could be examined to whatever had been routinely collected as part of the original UK Biobank cohort. We acknowledge that access to transport is a key determinant of active commuting, particularly for use of public transport. Access to a bike is also a key determinant of cycle behaviour. However, in the current population, the majority of individuals who used active commuting modes walked to work and this does not require any access to transport but is likely to be affected by the local environment. Therefore this is what we aimed to explore. A sentence has been added to the discussion section which identifies this limitation: "We were unable to examine the association of access to transport as a correlate of active commuting but it can be hypothesised this would correlate strongly with the transport method chosen".

3. The differentiation of active commuting measures may need further explanation. How do these differ?

We acknowledge that the differentiation of active commuting methods requires further explanation. We have added more detail to the methods section to explain this in more detail.

(1) car or car and public transport; why have these two been merged together?

Participants in UK Biobank were asked to respond to the question 'what types of transport do you use to get to and from work?' They were able to respond with more than one transport options. Therefore we had to make a decision on how to categorise people who reported more than one transport mode. The decision was taken to keep the category for active commuting as clean as possible, therefore only including people who only reported active commuting. This may underestimate the true impact of the behaviour but it does mean we can have confidence that any observed associations between active commuting, physical activity and sedentary behaviour are reflective of true associations and not an artefact of public transport or car use.

(2) public transport and mixed; what does mixed include?

In this case mixed transport refers to those who reported both public transport and active commuting or those who reported car/motor transport and active commuting.

and where is the option of car with cycling/walking?

This is included within the mixed category.

Although we recognise there is a limitation in using mixed categories and does not allow us to examine differences between single transport modes and mixed transport modes, the question used allow for a variety of responses to be given. Sample size did not allow for us to examine each different response variety and therefore the 3 categories used were a compromise. They are also similar to the categories used in previous research with the UK biobank cohort. The limitations of the categories used are discussed within the limitations section of the discussion.

The above may need to be addressed as additional limitations to the research.

Reviewer: 2

Abstract

Please add "UK Biobank"

Participant line, clarify type 2 DM

The abbreviation for AC is not spelt out in the abstract not given; do you want a line for “exposure measures”?

These changes have all been done.

Introduction

1) While the introduction is accurate, the case for why the work has been done could be much clearer. It may also benefit from being a bit shorter and more focused. BMI is relevant (particularly to aetiology of diabetes), but does it require two paragraphs?

My personal view is that the strongest argument for undertaking this work is that people with T2DM tend to be less active, and increasing physical active can be an important part of their treatment. AT is likely to be a sustainable way for people to maintain relatively high levels of PA – the aim of this work was to understand whether (in people with T2DM) AT was associated with increased PA; and further to understand the correlates of AT (with a view to designing interventions).

Some changes have been made to the introduction in light of these comments. These include reducing the focus on BMI and making the aim more explicit.

2) The shift from active travel to active commuting (with paragraph 3) is not explained. They are different.

All mentions of active travel have been changed to active commuting.

3) First line, the phrase “driven by” – may imply causes, and given this study is about people *with* diabetes, it is not informative in terms of aetiology

This has been changed to ‘associated with’

4) 2nd para – what proportion of patients with T2DM meet guidelines according to this data source, in your sample the prevalence looks quite high (>50%) with on average well over 150 minutes per week. In the Early Actid Study (referenced), participants spent an average of 25.4 ± 18.9 minutes of the day engaged in objectively measured MVPA. The values reported in the current sample are higher but probably reflect differences in the mode of measurement with self-report techniques prone to overestimation. As stated in the discussion it would be interesting to use the accelerometry data in Biobank to explore this in more detail; however, it would impact on sample size. In light of these comments we have changed the sentence in the introduction to read:

Fewer adults with T2DM achieve UK Government recommendations of at least 150 minutes moderate-to-vigorous physical activity (MVOA) per week.....

5) Line 34, suggest add “traffic” to clarify

This has been done

6) Ref 15 – are these responses based on ‘usual’ mode of travel (i.e. a single choice is forced)?

Would be helpful to clarify

The census questions asks ‘how do you usually travel to work?’. Participants are required to tick one box only, for the longest part of the journey. This has been clarified in the text.

7) Line 41 – it is not just **cross-sectional** evidence, you later cite longitudinal, perhaps observational

I have removed the word ‘cross-sectional’.

8) Page 5, line 15 – what are good traffic conditions, do you mean low traffic volumes? Or slow speed?

The evidence suggests that low traffic volumes and traffic calming measures may help to promote active commuting behaviours. This has been made more explicit in the introduction.

9) Page 5, Lines 11-18, is there any reason why these findings might be different in people with T2DM?

There is no evidence to suggest that these findings may be different in people with T2DM. However, this population is often characterised by low PA and high sedentary time. Much of the research

conducted to date has examined correlates of active commuting in healthy and active populations. We hypothesised that rates of active commuting would be low in people with T2DM and it may be that the correlates are different to the general population.

Methods

10) Line 44, please insert a space after 9.2

This has been done

11) Study sample, is it worth considering a flow diagram (item 13c on the STROBE statement)?

A flow diagram has been added.

12) Active commuting measures – it might be helpful to spell out, perhaps in a table, what these different groups are (i.e. which responses go into which groups), for those who are less familiar with the questionnaire. Would labels like: motorised travel; motorised travel plus active travel; active travel only be more descriptive?

We thank the reviewer for their comments and appreciate that there is some confusion in the categorisation of AC behaviours. We do not feel that a table is necessary but have attempted to make the text description more clear as follows:

In order to answer the two study objectives, two exposure variables were derived: 1. A three-category exposure variables; (1) motorised travel: car or car and public transport, (2) motorised travel plus active commuting; public transport and mixed (car and walk/cycle; public transport and walk/cycle); (3) AC; walk or cycle only and 2. A binary variable (1) No active commuting; (2) Active commuting (AC: walk or cycle only). Only participants who exclusively reported walking or cycling behaviour were included in the active commuting categories.

These category labels have also been updated in the tables and relevant sections.

13) Physical activity; I wonder whether a statement along the lines of “From these questions, we derived three measures of physical activity...”

This has been done.

14) Is total physical activity the best or usual term – it is a measure of physical activity energy expenditure (i.e. product of duration and intensity). Total PA doesn't quite feel right because you have only summed moderate, vigorous and walking (with walking often being moderate in intensity or close to)

In the IPAQ Guidelines for data processing, the continuous score is referred to as 'Total MET-min/week'. Therefore I have changed the variable name throughout.

15) Line 35 – should the units be MET-minutes (not MET/minutes)?

This has been done.

16) MVPA – would it be clearer to call this duration of MVPA (in part to distinguish from total PA)

As we have now changed the variable name for the total PA, we consider it reasonable to keep this as MVPA. The duration (minutes/week) is expressed in the data table.

17) Environmental correlates. Why were these measures chosen? Many of the measures seem sensible to consider, but I am less convinced about air pollution (of which people are largely unaware and may be correlated with other measures like density)

Where possible, the environmental correlates were selected on the basis of what prior research had indicated as being correlated with commuting behaviour (Stewart et al, 2015; Frank et al, 2007). One study conducted in the US found air quality to be associated with active commuting (Fan et al, 2014 <http://www.sciencedirect.com/science/article/pii/S1353829214001440>). It can also be hypothesised to be a proxy measure of congestion and traffic conditions. This reference has been added to the manuscript. There were some additional restrictions placed on the environmental measures in terms

of what had been collected by Biobank.

Table 1

18) For deprivation, could you please indicate the most and least deprived quintile

This has been done

19) Most of the numbers are % - could this be indicated please

This has now been indicated in the table and a sentence has been added to the statistical analyses section which reads 'continuous variables are displayed as mean and standard deviation and frequencies are used for categorical variables.'

20) Commute frequency – per week?

Yes this variable is number of outward journeys per week. This has now been indicated in the table.

21) Distance for home – mean or median?

This variable is expressed as a mean.

22) Please clarify what traffic density means; does this refer to the road on which the person lives?

In this instance traffic density refers to the average total number of motor vehicles per 24 hours on the nearest road based upon local road network measured in vehicles per day. This has been explained in the methods using the sentence:

Traffic intensity is measured as the average total number of motor vehicles per 24 hours on the nearest road.

Table 2

23) MVPA, suggest add label (minutes)?

This has been done

24) MET-minutes or MET minutes; suggest should be consistent with text; and is this per week or per day?

This has been done

25) Total sedentary time, please indicate is this per day?

This has been done

26) Could you please add the details of the models as a foot note to the table, i.e. the variables you
The models are explained in the footnote of the table.

Table 3

27) Why did you not adjust of employment and environment characteristics in a single model?

With the exception of population density in the males, none of the environmental correlates were shown to be associated with active commuting. Therefore we did not feel it was necessary to put both employment and environmental characteristics into a single model.

Results

28) Page 11, para 3: the values quoted are positive but the values given in bracket (as the 95% CI are negative), please review

The values are quoted as positive but the word 'fewer' has been used to indicate these represent a negative value. The paragraph has been changed to make it more explicit with negative signs included.

29) Page 11, line 36 – Table *3*

This has been done

30) Line 37, suggest add the p-value for the interaction term

This has been done.

31) Page 12, line 54 – "overwhelming" feels like an odd word; some other variables still remain

significant so whilst distance has a very strong association, other associations do still persist
This has now been changed to read 'strongest correlate'.

Discussion

32) Line 25, "higher PA" – please clarify MVPA or total PA

This has been clarified

33) Line 32, effect size or association. Observation studies (particularly cross-sectional ones), cannot demonstrate an effect.

This has been clarified

34) Lines 55-59; I do not see how these measures can overcome the problem of a long commuting distance.

I agree that these measures may not totally overcome the issue of a long journey; however, they may help to encourage individuals to choose an active commute mode for part of the journey, for example the provision of bikes for hire at train stations.

35) Page 14, line 1; could you elaborate a little further on why public transport use was associated with little or no effect on MVPA? Are people who undertake PT unlikely or not able report some of the walking associated with PT in the questionnaire used? Did you look at the descriptive stats by commuting category? In analyses I have done, those who use the public transport are much more likely to live further from home and/or have long duration of commutes (and thus it seems plausible that they may have to reduce other activities to compensate – e.g. recreational MVPA). For this reason comparing public transport users (depending on the characteristics of your sample) with car users may not be a 'fair' comparison if car commuters live closer to work.

We hypothesise that one reason for the lack of association seen is due to limitations of using self-report measures of physical activity. Walking behaviours in this context may be difficult to recall. It is also likely due to differences in the physical activity demands of different modes of public transport being wildly different, e.g. taking the tube is associated with walking behaviour whereas sitting on a train for a long commute may not be. Biobank (to my knowledge) does not collect data on the type of public transport used.

I have expanded this paragraph to include some of the limitations of our measure of public transport.

Conclusion

This might benefit from being more closely aligned to the study. I would consider dropping the statement "Furthermore, AC has the dual benefit...". Something along the lines of "strategies to increase active commuting in adults with T2DM may increase levels of physical activity, and thus contribute to reducing the burden of this disease", feels more appropriate. Improve the environment sounds too vague – and your work only looks at correlates.

This has been done.

VERSION 2 – REVIEW

REVIEWER	Oliver Mytton University of Cambridge, UK
REVIEW RETURNED	05-Jun-2017

GENERAL COMMENTS	Abstract AC can you clarify further please. Something like "reporting only undertaking walking or cycling to work". This is important – see my later comments. Methods Item 12: Thank you I think this is clearer. The titles you use in the text do not match the titles in the Table 2.
--

	Somewhere it would also be helpful to clarify that your three-category variable is used for objective one, and your two category variable for objective two (the text under statistical analysis just refers to correlates/association with AC) Item 17: A comment or statement about how/why you choose the variables for inclusion in your model would be helpful. Presumably also you were limited by what data was routinely available in Biobank (which is probably quite a serious limitation, in terms of having the data you might want)? I am still unconvinced about air pollution. The Fan paper was based in US urban areas (your sample is both urban and rural), was ecological and only had one significant association (at $p < 0.10$) for cycling. It seems either high air pollution \square deterrence of active commuting \square low AC; or it seems high air pollution is marker for congestion \square high AC. Your results showed no significant association – could be picked up in the discussion. Item 22) density (in table) or intensity (in text)? Table 2: could you please clarify in the footnote that model 1 is unadjusted (if that is correct) Table 3: title should state commuting not travel Item 28) I still find this confusing. I would either remove the number from the text and have “change -2.2, 95%CI -2.9 to -1.9” in brackets or remove the negative sign from the brackets. My comment about ‘aetiology’ and the words ‘primarily driven’ were about this being a paper that is concerned with people who already have type 2 diabetes Item 31) Overwhelming had not changed in the text (it had changed in the discussion, but not the results) Discussion “The results for public transport were mixed” – please clarify mixed. Your measure of AC (only walking/cycling to work) will have forced such a strong correlation for distance. 10 miles to work would take 3 hours (six hour round trip) of commute time if undertaken by foot. If your measure of active commuting was any active commuting on the journey to work (i.e. public transport + a walk to the office) then it is at least physically possible for people to undertake active commuting and still live far from work, and the strength of the association (likely) less. Your study hasn’t addressed the environment correlates of this pattern of active commuting (the pattern that is the only realistic pattern of active commuting for people who live beyond 3-5 miles from work) – this should be acknowledged in the discussion. See also comments above about measures of environment available in UK Biobank.
--	--

VERSION 2 – AUTHOR RESPONSE

Abstract

The abstract has been updated to include reference to ‘reporting only walking or cycling to work’.

Methods

The table has been updated to match the titles in the text.

This is explained in the methods section (active commuting measures where the following description is:

In order to answer the two study objectives, two distinct exposure variables were derived: 1. a three-

category exposure variable; (1) car or car and public transport; (2) public transport and mixed (car and walk/cycle; public transport and walk/cycle); (3) AC: walk or cycle only and 2. a binary variable ((1) No active commuting; (2) Active commuting (AC: walk or cycle only)).

Item 17: These environmental variables were mostly selected on the basis of data availability and following discussion with a colleague who has an interest in air pollution and environmental correlates of physical activity behaviour. As this was a student MSc project we were time limited and could not wait for further derivation of environmental variables. I have included a statement on this. These correlates were selected on the basis that they were available on an individual level and may help to inform further analysis of environmental correlates of commuting behaviour.

Item 22 – I have updated the text to read traffic density

Table 2 has been updated to include a footnote.

Table 3 has been updated to say active commuting.

Item 28, I have removed the negative signs from the brackets.

Item 31 – I have updated in the results as well as the discussion section.

Discussion

I have updated to include the following sentence;

In the current study, the results for public transport were equivocal with no apparent observations with physical activity observed, This contrasts with previous research (17) and may be due to an inability to differentiate between the different types of public transport taken and the public transport category also including people who reported mixed modes of motorised travel and active commuting.

In terms of air pollution I have included a statement in the discussion to acknowledge the lack of inter-individual variation in the pollution data available and how this may explain the lack of association seen:

The final aim of the study was to explore the social and environmental correlates of active commuting behaviour. Those who lived in a rural area were less likely to use active modes of commuting, but no other environmental variables, such as traffic density were shown to be associated.

This may be due to an overall lack of variation in the environmental variables and suggests a more diverse sample may be required. For example, there is a lack of inter-individual variation in the measure of air pollution which may explain the lack of association seen.

I acknowledge the comment regarding the patterns of AC included in the current analysis and have tried to address these limitations by including the following paragraph in the discussion:

Furthermore, we used a crude measure of active commuting (walking or cycling only) and were therefore unable to explore the role of active commuting as a contributor to the overall journey. For those individuals who live beyond 3-6 miles from work are unlikely to solely use active commuting modes to travel to work and therefore the role of active commuting as a component of the overall journey should be explored in future research.